# Photo-Isomerization Kinetics of Azobenzene Containing Surfactant Conjugated with Polyelectrolyte

**DOI:** 10.3390/molecules26010019

**Published:** 2020-12-22

**Authors:** Anjali Sharma, Marek Bekir, Nino Lomadze, Svetlana Santer

**Affiliations:** Institute of Physics and Astronomy, University of Potsdam, 14476 Potsdam, Germany; anjali.sharma@uni-potsdam.de (A.S.); marek.sokolowski@uni-potsdam.de (M.B.); nlomadze@uni-potsdam.de (N.L.)

**Keywords:** azobenzene, photo-sensitive surfactant, photo-isomerization kinetics, poly(acrylic acid, sodium salt)

## Abstract

Ionic complexation of azobenzene-containing surfactants with any type of oppositely charged soft objects allows for making them photo-responsive in terms of their size, shape and surface energy. Investigation of the photo-isomerization kinetic and isomer composition at a photo-stationary state of the photo-sensitive surfactant conjugated with charged objects is a necessary prerequisite for understanding the structural response of photo-sensitive complexes. Here, we report on photo-isomerization kinetics of a photo-sensitive surfactant in the presence of poly(acrylic acid, sodium salt). We show that the photo-isomerization of the azobenzene-containing cationic surfactant is slower in a polymer complex compared to being purely dissolved in aqueous solution. In a photo-stationary state, the ratio between the *trans* and *cis* isomers is shifted to a higher *trans*-isomer concentration for all irradiation wavelengths. This is explained by the formation of surfactant aggregates near the polyelectrolyte chains at concentrations much lower than the bulk critical micelle concentration and inhibition of the photo-isomerization kinetics due to steric hindrance within the densely packed aggregates.

## 1. Introduction

Azobenzene-containing surfactants have proved to be very important in the recent years due to their remarkable ability to reversibly change their molecular properties when exposed to light of different wavelengths [1,2,3,4,5,6]. The reason for this is the photo-isomerization of azobenzene molecules from a stable *trans* to a metastable *cis* state accompanied by variation in the polarity, i.e., the *trans* state is less polar in comparison to the *cis* isomer [7]. This allows for reversible control over the solubility, critical micelle concentration (CMC) and interfacial energy of such surfactant molecule, as well as the strength of interactions, simply with optical stimuli [8,9,10]. For example, one can make polyelectrolytes and DNA molecules photo-sensitive by preparing supramolecular complexes with azobenzene-containing surfactants [11,12,13,14,15,16] It is also possible to make polyelectrolyte brushes photo-responsive by loading them with oppositely charged photo-sensitive surfactant, which allows for a structuring of the brush by optical stimuli [15,16,17].

To explain the dynamic response of conjugated soft objects upon illumination, knowledge about the isomerization kinetics of photo-sensitive surfactants and the isomer composition at the photo-stationary state is of a fundamental importance [18,19,20,21,22,23]. For instance, when charged microgels interact with photo-sensitive surfactant, one can change the microgel size between a swollen and collapsed state by adjusting the concentration of the *trans* isomers in solution during irradiation with an appropriate wavelength, while keeping the absolute surfactant concentration constant [24,25]. The different ratios of *trans* and *cis* isomers is achieved by changing the irradiation wavelength, for instance, under exposure to light of a 455 nm wavelength, the *trans*/*cis* ratio is 60/40%, while upon irradiation with UV light (365 nm), it is 5/95% [26]. The size response takes place during the dynamic exchange of two isomers between the bulk and the microgel due to different interaction affinities of the two isomers. The *trans* isomer tends to accumulate in the negatively charged pores of the particles, while the *cis* isomer prefers to stay in solution. A similar process changes the inter-particle interaction potential in an ensemble of rigid mesoporous colloids, resulting in reversible aggregation or separation of the particles under irradiation with light [27]. Here, when exposed to UV light, the formed *cis* isomers within the pores readily diffuse out of the particles and generate an excess concentration near the colloids’ outer surface, ultimately resulting in the initiation of diffusio-osmotic flow [28,29,30]. The direction of the flow depends strongly on the dynamic redistribution of the fraction of *trans* and *cis* isomers near the colloids due to different kinetics of photo-isomerization within the pores as compared to the bulk. Similarly, in the case of polyelectrolyte-surfactant complexes depending on the ratio of the *trans/cis* isomers dynamically changing under irradiation, one can find the polyelectrolyte chain either in a coil or globule state, as shown previously [31]. With the examples discussed above, we demonstrate that the knowledge of the photo-isomerization kinetics of the azobenzene-containing surfactant conjugated with the oppositely charged object is an integral part of the understanding of the light-triggered structural response.

Here, we report on photo-isomerization kinetics of azobenzene-containing surfactants complexed with poly(acrylic acid) sodium salt (PAA) in aqueous solution as a function of several parameters, such as polymer and salt concentration, as well as the wavelength and intensity of applied irradiation using a similar model as reported elsewhere [26].

## 2. Results

The chemical structure of azobenzene-containing surfactants and the UV-Vis absorption spectra of the polymer-surfactant complex are shown in Figure 1. The *trans* isomer has a characteristic absorption band (π-π* transition) with a maximum at 351 nm. The spectrum of the *cis* isomer is characterized by two absorption bands with maxima at 313 nm (π-π* transition) and at 437 nm (n-π* transition). The lifetime of the *cis* isomer in the dark or under illumination with red light of *λ* = 600 nm is ~40 h at 20 °C, while the photo-isomerization from the *cis* to *trans* state under irradiation with blue light (*λ* = 455 nm) takes place much faster, as discussed in the following section.

The photo-stationary state with fractions of *trans* and *cis* isomers of 60% and 40% at 0.1 mM surfactant concentration, respectively, is reached after a certain time of irradiation, which is dependent on the light intensity [26]. Under UV illumination (*λ* = 365 nm) at the photo-stationary state, the surfactant molecules are predominantly in the *cis* state, at a fraction of 95%. The UV-Vis absorption spectra of the pure polymer (black line) and pure surfactant (blue and green lines), along with that of the PAA-surfactant complex are shown in Figure 1c. When the complex is not irradiated, i.e., ~100% of surfactant molecules are in the *trans* state (red curve in Figure 1c), there is a blue shift of the adsorption maximum from 353 nm (blue curve in Figure 1c) to 341 nm, indicating surfactant aggregation in the complex into micelles even when far away from the bulk CMC [27]. After exposure to UV light (*λ* = 365 nm) the amount of *cis* isomers increases to 80%, and the spectrum is similar to a pure surfactant one (compare green and orange curves in Figure 1c).

The time-resolved change of the *trans* isomer concentration in the polyelectrolyte complex normalized by the initial concentration, *c*_T_/*c*_T,0_, is shown during irradiation with UV (Figure 2a) and blue light (Figure 2b). The intensity of the irradiation is kept fixed at 1 mW/cm^2^ for both wavelengths. Under UV light, the concentration at equilibrium for the *trans* isomers in the polymer-surfactant complex is much lower (*c*_T,eq_/*c*_T,0_ ~ 0.1) compared to that under blue light (*c*_T,eq_/*c*_T,0_ ~ 0.8). In the case of pure AzoC_6_ (without any polymer), *c*_T,eq_/*c*_T,0_ yields for UV exposure a value of ~0.05 and for irradiation with blue light, *c*_T,eq_/*c*_T,0_~0.6 (Figure 2d). Thus, at a photo-stationary state, the amount of *trans* isomers is larger in the complex with polyelectrolyte compared to the bulk value. Consequently, values of the rate constants of PAA-AzoC_6_ complexes, *k*_TC_ and *k*_CT_, are lower compared to the pure AzoC_6_ (as calculated using Equation (4), see Figure 2e,f). The characteristic time of photo-isomerization, *τ*, is calculated using Equation (3), which essentially stems from the exponential fitting of the data in Figure 2a,b. The *trans* isomer concentration, *c*_T_/*c*_T,0_, follows a simple exponential decay approaching equilibrium, due to the pseudo first-order relation of the isomerization rate.

One can clearly observe that the value of the decay time of photo-isomerization is considerably smaller than that of the PAA-surfactant complex (Figure 2c). This can be explained in the case of irradiation with blue light (blue triangles and dashed blue line in Figure 2c) in the following way: when small amount of polyelectrolyte is added (green area in Figure 2c), the majority of the surfactant is bound to a polymer chain, forming aggregates and the photo-isomerization kinetic slows down due to steric hindrances. Since the measured photo-isomerization time comprises the contribution of bound and free surfactant, with increasing polymer concentration (in the green area), the photo-isomerization becomes even slower since less free surfactant is present in solution up to a certain saturation point where the equilibrium between bound and free surfactant is achieved. Under irradiation with UV light (pink rectangles in Figure 2c) after the first slowing down of the photo-isomerization process (as in the case of blue irradiation), the value of decay time decreases with a further increase in polymer concentration.

This might be defined by the fact that when the UV light is applied, almost all surfactant molecules in the bulk switch to the *cis* state and, to keep the equilibrium, the bound surfactant in the *trans* state leaves the polymer chain, so that the contribution of the photo-isomerization kinetic from the sterically hindered surfactant decreases. We exclude the change of recorded intensity due to the precipitation of the PAA-surfactant complex by detecting solution turbidity for the largest concentration of PAA (15 wt%) (as described in Appendix A). The absence of complex precipitation is also supported by the values of the charged ratio, Z, between the polyelectrolyte and surfactant charges (see Section 1 and Appendix A). Indeed, the amount of the surfactant is still not enough to compensate all polyelectrolyte charges, thus stabilizing the PAA-AzoC_6_ complex via overcharging with PAA moieties.

In the next step, we show how the isomerization kinetic of PAA-AzoC_6_ complexes (recorded for the largest concentration of PAA of 15 wt%) depends on the irradiation wavelengths (365, 455, 490 and 532 nm), keeping the intensity constant at 1 mW/cm^2^ (Figure 3). Under illumination with UV light, the ratio *c*_T,eq_/*c*_T,0_ is the lowest (Figure 3a), and increases with increasing wavelength. This is a similar trend as in the case of pure surfactant, as reported in our previous publication [26]. The absolute values of *k*_TC_ and *k*_CT_ for the PAA-surfactant complex differ from those of the bulk (Figure 3b). For all wavelengths, we observe a reduced photo-isomerization kinetic in complex, with an equilibrium shift towards an increasing *trans* isomer concentration in a steady state. This results from a reduced forward reaction due to the slowing of the reaction rate from *trans*–*cis* due to steric hindrance inside a micelle and the strong interaction between the surfactant (*trans*) and polymer, as discussed in detail later. We also calculated the quantum efficiency for the isomerization, yielding a value of *Φ* = 0.12 for 365 nm and decreasing with increasing wavelength (see Appendix A). Apparently, isomerization is most efficient when the molecules are exposed to UV light.

The kinetic of surfactant photo-isomerization in a complex with PAA (*c*_PAA_ = 15 wt%) also depends on the intensity of the applied irradiation (Figure 4). Thus, the decay time decreases with increasing intensity (Figure 4c), but the steady state concentration reaches the same values (Figure 4b). The values of *k*_TC_ and *k*_CT_ of the polymer-surfactant complex for both wavelengths do not vary significantly with intensity (Figure 4d), as in the case of pure surfactant [28].

We should mention that the photo-isomerization kinetic of the surfactant in a complex with PAA does not depend on the molecular weight of the polymer, as shown for two cases, MW = 0.5.104 g/mol and MW = 25.104 g/mol (see Appendix A), since the polyelectrolyte size has a minor influence on the degree of ionization [32].

The decrease in the isomerization rate of the surfactant in the complex compared to a bulk value may correlate with the ionic strength of the solution as it shifts the critical micellar concentration (CMC). It has been reported previously that the rate of isomerization in micelles is 80% slower [26]. Therefore, we measure the isomerization kinetics of surfactant in the presence of two salts, NaBr and NaAc. The latter is the most structurally similar salt compared to the monomer unit of the PAA polymer. We have previously found [33] that at onset of the CMC, a blue shift of the absorption peak of the *trans* isomer is observed from *λ* = 351 nm up to *λ* = 340 nm. In Figure 5, the peak maximum *λ*_max_ is plotted as a function of the ionic strength. The strongest blue shift is observable for NaBr, which sets at 20 mM, indicating the micellization of 0.1 mM surfactant, while for the NaAc, the CMC is at ~95 mM. The difference is attributed to counter-ion-specific interactions, since in the case of NaBr and AzoC_6_, the anion is the same (Br^−^). For comparison, we have added to the plot the shift of the maximal absorption peak of the surfactant in the presence of NaCl, where the onset of micellization is at ~45 mM (as found in the literature) [33]. In the presence of PAA, the micellization of the 0.1 mM AzoC_6_ surfactant concentration begins at approximately 30 mM ionic strength concentration mediated from PAA and corresponds to a mass concentration of 0.68 wt%. Note that the degree of ionization is assumed to be 0.5, estimated from titration measurements (for details, see Appendix A).

In Appendix A, we summarize the time-resolved *c*_T_/*c*_T,0_ of the salt-AzoC_6_ mixture (I = 1 mW/cm^2^) with concentrations varying from 4 to 780 mM for UV and blue irradiation, respectively (calculated using data presented in Appendix A). The values for *k*_TC_ and *k*_CT_ depicted in Figure 6a,b as a function of the ionic strength *c*_S_ during irradiation with UV and blue light are calculated using the decay time and *trans*/*cis* ratio at a steady state (Appendix A). It is interesting to note that there is a strong correlation between the rate constant reduction and micelle formation (see Figure 6). Values for *k*_TC_ are constant at a low salt concentration and decay, starting from a critical salt concentration where the micellization takes place (see Figure 5). Starting from a critical salt concentration (for NaBr at 20 mM and for NaAc at 95 mM), the number of micelles increases with concentration, resulting in a further decrease in *k*_TC_ (Figure 6). This is in good agreement with the previously reported results, where an increase in surfactant concentration above the CMC results in a continuous drop of the *k*_TC_ value [26]. For both wavelengths in Figure 6a,b, values for *k*_CT_ are almost constant over the total ionic strength range studied in this work due to the fact that *cis* isomers are not in the aggregated state (CMC of *cis* in bulk is approximately eight times larger than in the *trans* state).

In the schematic illustration (Figure 7), we posit the cause of the reduction in photo-isomerization kinetics of the surfactant in complex with polyelectrolyte due to steric hindrance in the micelles formed at the polyelectrolyte. The latter is correlated with a CMC shift caused by the aggregation of the surfactant molecules near/at the poly-ion. [34] The critical micellar concentration (CMC) nearby and inside the coil of the poly-ion may be shifted to a lower concentration similar to what was reported in the literature for the interaction with negatively charged microgels [35]. The surfactant molecules tend to be in two phases: aggregated in a complex (values for *k* are low) and free in solution (*k* values are larger).

## 3. Materials and Methods

### 3.1. Light-Responsive Surfactant

The azobenzene-containing trimethylammonium bromide surfactant (C_4_-Azo-OC_6_TMAB, abbreviated in this work as AzoC_6_) was synthesized as described elsewhere (Figure 1a) [33].

Samples were prepared by diluting a 10 mM aqueous surfactant stock solution to the desired surfactant concentration of 0.1 mM and desired salt or polymer concentration with Millipore water, aqueous salt (*c* = 1500 mM) and PAA stock solutions (*c*_PAA_ = 20 wt%).

Poly (acrylic acid) partial sodium salt, NaBr and NaAc were purchased from Merck KGaA, Darmstadt, Germany and used without further purification. A stock solution of different salts of 1105 mM was prepared and diluted to the required concentrations, ranging from 4 mM to 780 mM.

### 3.2. Characterizations

Time resolved UV-Vis measurements were performed with a commercial Cary 5000 UV-Vis-NIR spectrophotometer instrument (Agilent Technologies, Santa Clara, CA, USA). A 1 cm thick rectangular quartz cuvette, transparent in all directions (Helma Analytics, Berlin, Germany), was filled with 2 mL of aqueous solution and sealed in order to keep the concentration constant during the measurement. An LED lamp (Thorlabs GmbH, Lübeck, Germany) (perpendicular to monitoring beam) illuminated the total volume of the sample holder for fixed wavelengths *λ* = 365 nm, *λ* = 455 nm, *λ* = 490 nm, *λ* = 530 nm. Absorption spectra of the surfactant showing the different concentrations of *trans*/*cis* species at different illuminations can be seen in Appendix A. Before each measurement, the intensity of the light source was measured using a commercial S170C power meter (Thorlabs GmbH, Lübeck, Germany). A low-pass filter (10SWF-400-B, Newport Corporation, Darmstadt, Germany), cut-off from 400 nm and higher was placed in the beam path, between the holder and detector (see Figure 1d and Appendix A). Time-resolved absorbance was recorded at *λ* = 376 nm (monitoring beam intensity, I = 0.02 mW/cm^2^) until it reached the stationary state.

The concentration of the surfactant was calculated from the initial value of absorbance based on the knowledge of the adjusted concentration from a stock solution. We assumed that the total surfactant concentration was equal to the *trans* isomer concentration, *c*_T,0_, at the initial time of illumination. The absorbance was monitored at a wavelength of 376 nm to calculate the ratio of *trans* isomers *c*_T_(t)/*c*_T,0_. This particular wavelength was selected as the absorption of the *cis* isomer is minimal at this wavelength. *c*_T_(t)/*c*_T,0_ can be calculated as follows with Equation (1):(1)$$\frac{c_{T}\left(t\right)}{c_{T,0}}=\frac{\left(\mathrm{Abs}-{\mathrm{Abs}}_{S}\right)}{\left({\mathrm{Abs}}_{0}-{\mathrm{Abs}}_{S}\right)},$$
where Abs is the absorbance of the photo-sensitive surfactant-polymer complex at 376 nm at any time and before irradiation Abs_0_. The value of the absorbance mediated from the scattering of PAA only, Abs_S_, is obtained by recording the absorbance of the pure polymer at the same wavelength for the same polymer concentration (see Appendix A).

### 3.3. Kinetic Model and Data Interpretation

The photo-isomerization kinetic of a pure surfactant in bulk has been previously described in detail elsewhere [29]. In short, the *trans* isomer concentration was calculated as a function of irradiation time using Equation (2):(2)$$c_{T}=\frac{k_{\mathrm{CT}}+k_{\mathrm{TC}}\cdot \exp\left(-\left[k_{\mathrm{TC}}+k_{\mathrm{CT}}\right]\cdot I\cdot t\right)\;}{k_{\mathrm{CT}}+k_{\mathrm{TC}}}\cdot c_{T,0},$$
with *k*_TC_ and *k*_CT_ being the isomerization rate constant and *c*_T_ and *c*_T,0_ being concentrations of *trans* isomers at any time and initial time (i.e., when the concentration of the *trans* species is ca. 100%), respectively. Exposing the surfactant solution to light leads to concentration fractions of *trans* and *cis* isomers being in equilibrium, where both rates are equal, i.e., d*c*_T_/dt = −d*c*_C_/dt and results in a rate constant ratio *X* between *k*_TC_ and *k*_CT_, *X*∙*k*_TC_ = *k*_CT_. Then Equation (2) can be transformed into Equation (3):(3)$$c_{T}=c_{T,\mathrm{eq}}\cdot \left(1-\exp\left(-\frac{t}{\tau }\right)\;\right)+c_{T,0}\cdot \exp\left(-\frac{t}{\tau }\right),$$
with characteristic time *τ* as shown in Equation (4):(4)$$\tau =\frac{1}{(1+X)\cdot k_{\mathrm{TC}}\cdot I},$$
with $X=\;\frac{c_{T,\mathrm{eq}}}{c_{T,0}}/\left(1-\frac{c_{T,\mathrm{eq}}}{c_{T,0}}\right)$ or $\frac{1}{X}=K_{\mathrm{eq}}$where *c*_T,eq_ is the concentration of *trans* isomers at a photo-stationary state.

## 4. Conclusions

We report on isomerization kinetics of light responsive azobenzene-containing surfactants (AzoC_6_) forming a complex with poly(acrylic acid) salt as a function of polymer concentration, irradiation intensity and wavelength, as well as ionic strength for two different salts, NaBr and NaAc. The photo-isomerization was studied by the time-resolved UV-Vis absorbance, where the irradiation of the samples with different wavelengths and intensities is introduced in situ during simultaneous recording of the absorbance. The photo-isomerization of the surfactant is different for PAA-AzoC_6_ complexes in comparison to AzoC_6_ in bulk, where the isomerization rate is generally lower in PAA-AzoC_6_ complexes and yields different values of the equilibrium concentration. The concentration fraction of the *trans* and *cis* isomers at the photo-stationary state for PAA-AzoC_6_ is generally higher than for pure AzoC_6_. We have found also that the photo-isomerization kinetics vary with light intensity and wavelength. This suggests that it applies to surfactant complexes with maybe any polymers but strongly to weak polyelectrolytes since a surfactant is a surface-active material and will always have a certain tendency to aggregate at interfaces and macromolecules in a certain fashion.

## Figures and Tables

**Figure 1 molecules-26-00019-f001:**
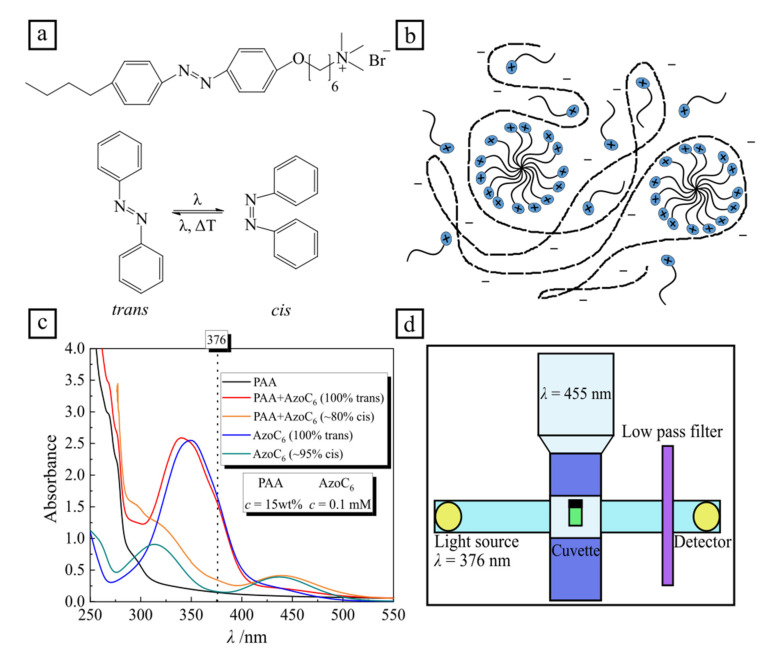
(**a**) Chemical structure of the azobenzene-containing cationic surfactant (AzoC_6_) and a schematic representation of azobenzene isomers. (**b**) Cartoon of the polymer-surfactant complex. (**c**) Absorption spectra: pure poly(acrylic acid) sodium salt (PAA) (*c* = 15 wt%, black line), PAA-surfactant complex with 100% *trans* isomers (red line) and ~80% *cis* isomers (orange line), pure surfactant (*c* = 0.1 mM) with 100% *trans* isomers (blue line), pure surfactant with ~95% *cis* isomers (green line). (**d**) Scheme of the experimental setup: continuous monochromatic light (*λ* = 376 nm) passing through the quartz cuvette is recorded by the detector in the course of irradiation with blue light, *λ* = 455 nm.

**Figure 2 molecules-26-00019-f002:**
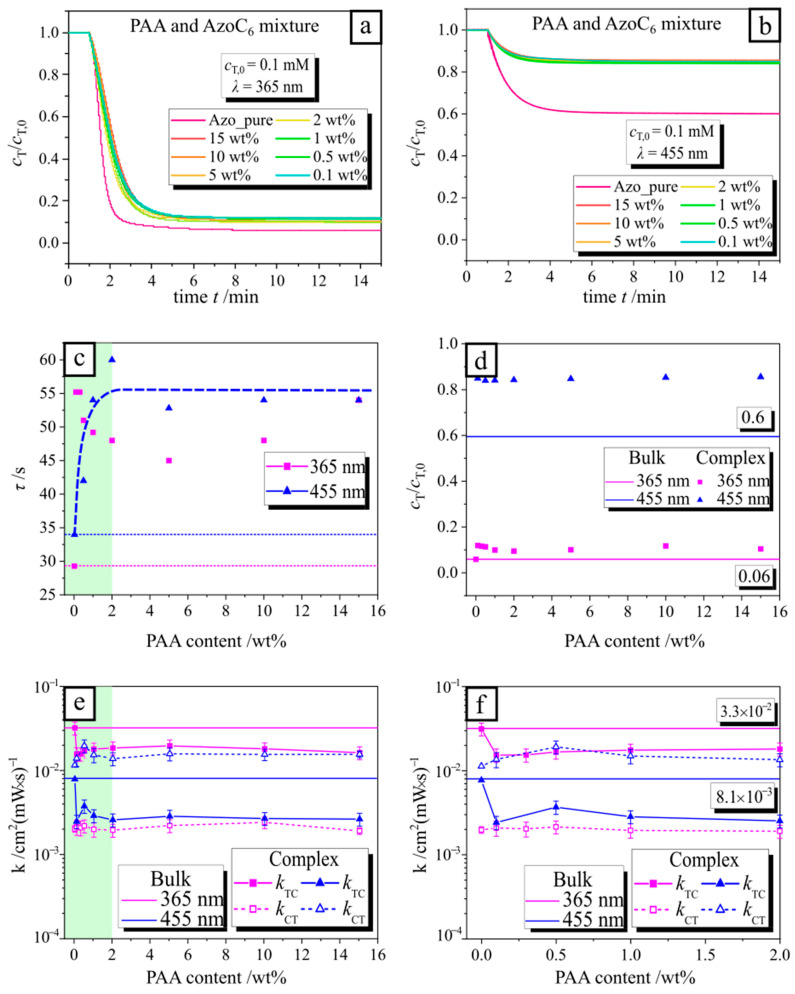
Time-resolved change in the amount of *trans* isomer, *c*_T_/*c*_T,0_, of the poly(acrylic acid) sodium salt and azobenzene surfactant (PAA-AzoC_6_) complex under irradiation with 365 nm (**a**) and 455 nm light (**b**). (**c**) Decay time for photo-isomerization as calculated by fitting the data from (**a**) and (**b**) using Equation (4). The dashed lines show the data for 0 wt% PAA (pure AzoC_6_). The green highlighted region illustrates a low concentration of PAA. (**d**) Fraction of *trans* isomer at a photo-stationary state. Data for PAA-surfactant complex is shown as dots, while the solid line represents the value without PAA. (**e**) Calculated rate constant from the decay time using Equation (4) for forward *k*_TC_ and reverse *k*_CT_ isomerization in PAA-AzoC_6_ complex. For a better view of the green marked area, a zoomed-in view from 0 to 2 wt% is shown in (**f**).

**Figure 3 molecules-26-00019-f003:**
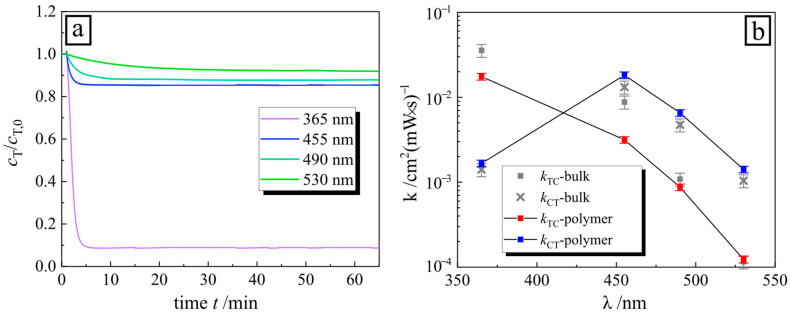
(**a**) Dependence of the *trans* isomer concentration normalized by the initial concentration*c*_T_/*c*_T,0_in poly(acrylic acid) sodium salt and azobenzene surfactant (PAA-AzoC_6_)complex (concentration of PAA, *c*_PAA_ = 15 wt%) on time during irradiation with light of different wavelengths: 365, 455, 490 and 530 nm. (**b**) Comparison between the rate constants of pure surfactant (*k*_TC_ as solid gray squares, *k*_CT_ as gray crosses) and in complex with polymer (*k*_TC_ as solid red squares, *k*_CT_ as solid blue squares).

**Figure 4 molecules-26-00019-f004:**
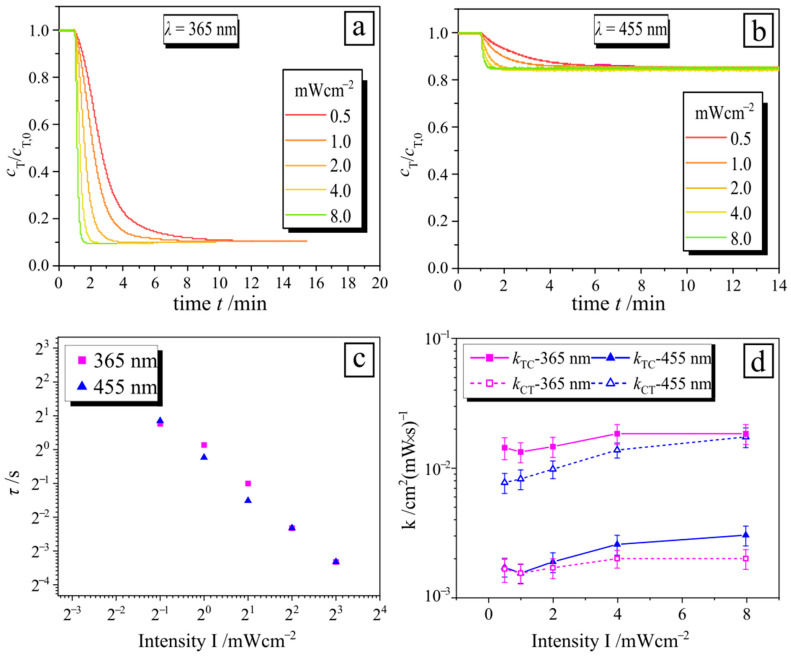
(**a**,**b**) Time-resolved *trans* isomer concentration normalized against initial concentration *c*_T_/*c*_T,0_ in the polymer-surfactant complex during irradiation with different intensities, marked on plots: (**a**) UV and (**b**) blue irradiation. (**c**) Photo-isomerization decay time as a function of irradiation intensity for two wavelengths: UV (pink squares) and blue (blue triangles). (**d**) Rate constants for the forward (*k*_TC_) and reverse reaction (*k*_CT_) of photo-isomerization of the polymer-surfactant complex for two wavelengths, as indicated in the legend.

**Figure 5 molecules-26-00019-f005:**
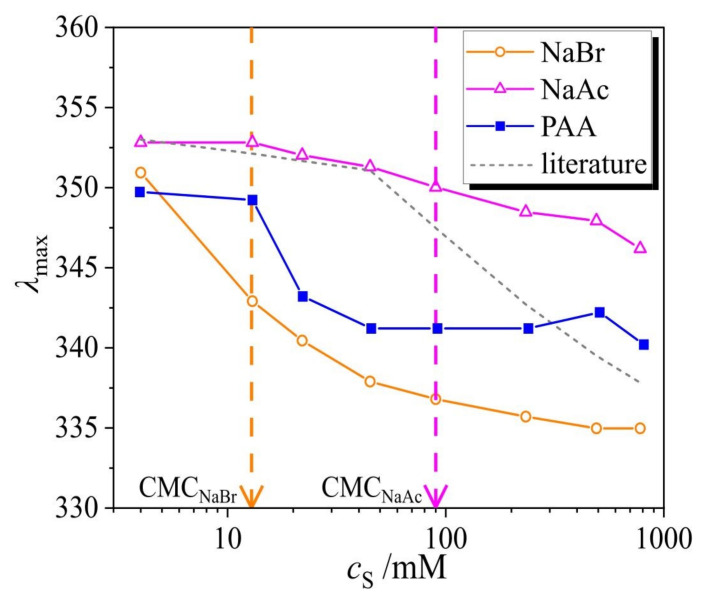
Dependence of the position of the peak maximum of the *trans* isomer, *λ*_max_, on the ionic strength *c*_S_ for two different salts, NaBr and NaAc, and the polymer PAA. The gray dashed line in the plot shows the shift in the presence of NaCl, as taken from our previous publication [27].

**Figure 6 molecules-26-00019-f006:**
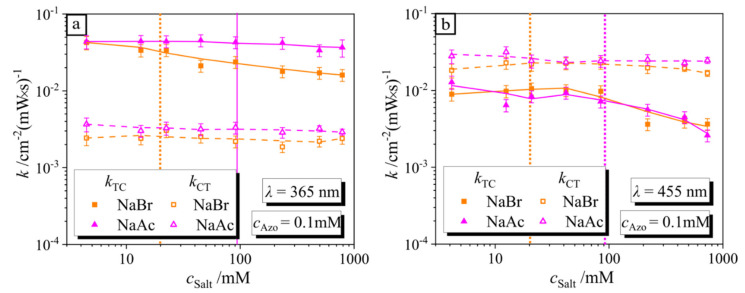
(**a**) Rate constants for the forward (*k*_TC_, solid points) and reverse (*k*_CT_, hollow points) reactions plotted against the concentrations of different salts depicted in the legends during irradiation with (**a**) UV and (**b**) blue light. The orange and pink dashed lines illustrate the critical micelle concentration (CMC) for NaBr and NaAc, respectively.

**Figure 7 molecules-26-00019-f007:**
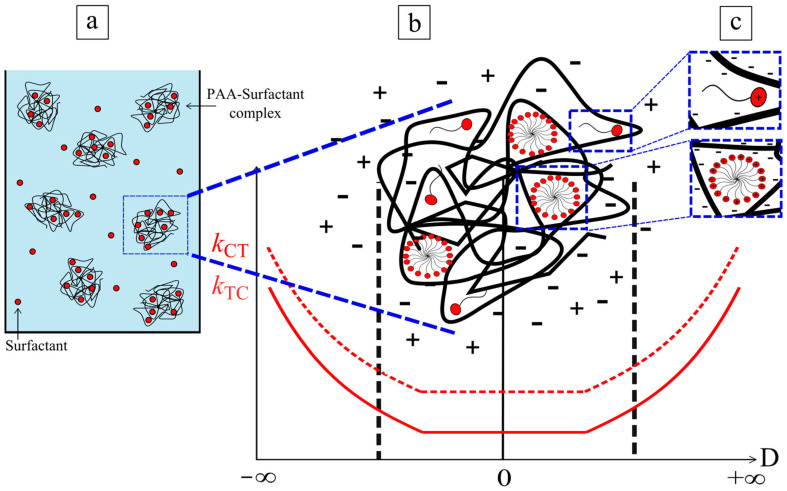
Schematic representation of the shift in the kinetics of isomerization in complex and in bulk.

## Data Availability

The data presented in this study are available in insert article or Appendix A here.

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
