# Peer review of "Photo-Isomerization Kinetics of Azobenzene Containing Surfactant Conjugated with Polyelectrolyte"

_molecules, 2020, doi:10.3390/molecules26010019_

Round 1
Reviewer 1 Report
The manuscript presents an interesting characterization study of photoisomerization of azobenzene containing surfactant in a polyelectrolyte solution. I find the study timely and reasonably conducted but at the same time I have fundamental questions about the setup, data and conclusions. I find the work publishable (with revision) but that some essential features of the system have been omitted. In particular
- PAA is a weak polyelectrolyte. What is its charge state? Does this remain same throughout the variation of salt and surfactants in the system? This influences interpretation rather drastically.
- What is the pH and is this controlled in any way?
- Pure Azobenzene surfactant system shows different relaxation time as any of the polymer containing samples, even the very low 0.1wt% polymer solution in Figure 2. Furthermore, increasing the polymer concentration from 0.1wt% to 15wt% does not have much effect on the relaxation. This is also visible in the rate constant etc. analysis. Why is this? It appears as if a very small amount of polymer changes the response but then the response saturates and crowding the system with the polyelectrolyte does not influence. I find this perplexing.
- Why do the Br and acetate ions differ in response? This relates to the ion charge distribution, hydration, and how the ions are binding with the PAA charge (and surfactant head groups). What can be deduced of the difference?
- I find the extension of the conclusions that this “begs to the suggestion that it applies to surfactant complexes with any polymers” such that I cannot agree. The studied polymer PAA is a polyelectrolyte. Furthermore, it is a weakly charged polyelectrolyte and the system contains ions. The findings are not general to any polymer. Please revisit the conclusions and especially this sentence with a bit more careful wording.
- How large are the missing error bars / error estimates in Figures 2-6? Do the fluctuations cover the differences or not?
Author Response
Dear Reviewer,
Thank you very much for the comments and suggestions. In the following we answer to them:
- PAA is a weak polyelectrolyte. What is its charge state? Does this remain same throughout the variation of salt and surfactants in the system? This influences interpretation rather drastically.
Our answer:
The degree of protonation has been measured and is reported in supporting information (Section 2, Figure S11 page S17 in Supporting Information). Since the Millipore water was used without further pH adjustment, the solution pH is around 6 and the degree of protonation 0.5 and is irrespective of molecular weight as reported in ref. 35.
The added surfactant changes the pH value between 6 and 6.8, which results to variation of the degree of ionization between 0.50 and 0.58 (see Figure S11 in Supporting Information), i.e. by 10%. This change alters the calculation (Section 1 eq. S1-S6) of dissolved salt concentration by 10% (which is in the range of the error bar) as the ionic strength resulting from the degree of ionization of α = 0.5 of PAA was assumed for the ionic strength calculation in Figure 5.
PAA was not mixed with any salt here. Thus, the degree of charge dependency as a function of salt is not relevant.
- What is the pH and is this controlled in any way?
Our answer:
The solution pH is at 6.
- Pure Azobenzene surfactant system shows different relaxation time as any of the polymer containing samples, even the very low 0.1wt% polymer solution in Figure 2. Furthermore, increasing the polymer concentration from 0.1wt% to 15wt% does not have much effect on the relaxation. This is also visible in the rate constant etc. analysis. Why is this? It appears as if a very small amount of polymer changes the response but then the response saturates and crowding the system with the polyelectrolyte does not influence. I find this perplexing.
Our answer:
we think, that at 2 wt% basically all surfactant is already in complex with the polyelectrolyte. Further increase of PAA content does not change significantly an amount of complexed surfactant. This may explain the saturation at 2wt% PAA content.
The complex leads to a confinement of surfactant molecules and ultimately to a reduced isomerization kinetics due to steric hinderance form trans to cis isomerization. (J. Chem. Phys. 152, 024904 (2020); https://doi.org/10.1063/1.5135913).
- Why do the Br and acetate ions differ in response? This relates to the ion charge distribution, hydration, and how the ions are binding with the PAA charge (and surfactant head groups). What can be deduced of the difference?
Our answer:
In this paper we do not study the salted solution of PAA. The results of the photo-isomerization kinetic of the complexes PAA with surfactant were compared with the salted surfactant. Since we did not perform any measurements for PAA as a function of ion type we don’t comment on it.
When comparing the influence of different salt on micellization of the surfactant, it is clear that the counterionic specific interactions with the head group alter the process significantlyas shown f.ex. in Advances in Colloid and Interface Science 146 (2009) 42–47, https://doi.org/10.1016/j.cis.2008.09.010). This in turn has a drastic effect on isomerization, as we also report in Figures 5 and 6.
We currently working on another manuscript to correlate isomerization and details of structural response of the surfactant under different salt solutions. Here we apply for the beam time in order to perform different scattering measurements. This kind of work defines a separate peace of study and will be reported in another manuscript.
- I find the extension of the conclusions that this “begs to the suggestion that it applies to surfactant complexes with any polymers” such that I cannot agree. The studied polymer PAA is a polyelectrolyte. Furthermore, it is a weakly charged polyelectrolyte and the system contains ions. The findings are not general to any polymer. Please revisit the conclusions and especially this sentence with a bit more careful wording.
Our answer:
We thank the referee for the critical revise and agree with the referee. Thus we updated the text.
Previous text: page 10 lines 289. “This begs to the suggestion that it applies to surfactant complexes with any polymers since surfactant is a surface active material and will always have a certain tendency to aggregate at interfaces.”
Corrected version: page 10, lines 289. “This begs to the suggestion that it applies to surfactant complexes with maybe for any polymers but strongly for weak polyelectrolytes since surfactant is a surface active material and will always have a certain tendency to aggregate at interfaces and macromolecules in a certain fashion.”
- How large are the missing error bars / error estimates in Figures 2-6? Do the fluctuations cover the differences or not?
Our answer:
The error bars are now added into the Figures 2-6. The fluctuations are now covering the difference. The errors were estimated from error propagation data and do not exceed 15% deviation from measured value.
Reviewer 2 Report
Dear Authors,
1- The introduction is good with 33 references. However, the line 31 must be rewritten. Indeed, if a variation of polarity in azobenzenic molecules is known between trans and cis form, the trans is non-polar only for 4,4'-symmetrical azobenzenes. For most of azobenzenic surfactants, we can only say that the cis-isomer is usually more polar than the trans one, due to its difference of conformation. In line 46 (ref 28) the authors should precise the wavelength called "UV light".
2- Concerning the results part, the authors claim variations of trans/cis ratios in percentage. However, it seems that no technical analyses were described for this quantification. Sometimes, thus, 100% of an isomer is claimed. Is this isomer crystallized? If yes, could the authors provide an analysis of this crystal? If no, the authors may add in supporting information their chromatographic and/or analytical spectra (HPLC with the two -trans and cis- wavelengths detector, or NMR sufficiently high in field).
Still in this part, what can the authors say about the cis-trans-cis-trans... isomerization sequences? Is there any change in the maximal absorbance of trans and cis isomers over isomerization cycles? Is there any change in the aggregation phenomenon?
Concerning Figure 1, how did the author to confirm that the percentage of pure surfactant with cis isomer (80% or 95%) was stable all along the time of analysis?
Concerning the Figure 2, one can have a doubt between irradiation wavelength (365 nm) and the analytical one (376 nm). The authors may precise it in the Figure 2 legend.
For the rest of the document, every physicochemical analyses are well-documented to lead to the conclusion that as for azobenzenes free in solution, kinetics are depending on wavelength and intensity, and to a valuable suggestion that surface-tension active material can aggregate at interface with polymers.
Author Response
Please, find our answer as attached file

Reviewer 3 Report
Santer and coworkers report on the isomerization of an azobenzene containing surfactant.
The study is well conducted and the results are clearly exposed.
I have just a few concerns:
- All the spectra shown in the text and in the SI are characterized by absorptions higher than 1.5. Have the authors checked that the traces they have recorded are in the Lambert-Beer regime?
- The authors should report the fitting of the original decays in the SI with the associated R^2.
- Considering that the authors already have all the decay traces related to the isomerization process, they should calculate the quantum yields of isomerization at different wavelenghts of irradiation. Comparing the kinetics by itself could be seen as a rather incomplete treatment of their analysis.
Some minor comments
- Figure 1c Absorbance should be unitless, it’s not expressed in wavenumbers
- Line 40 “is of s fundamental” correct the mistake
- Line 46 specify the wavelength for the UV light. Using, for example, 310 nm or 365 nm has quite a different effect in the photostationary distribution of azobenzene (especially for the unsubstituted one that has a maximum around 310 nm)
- Line 73 the scheme reports 490 nm but in the text 455 is reported. Also, in the picture it seems that you are isomerizing unsubstituted AB. Correct the figure adding the substituents of the photochromic core.
- Line 333 “Photoisomerization of Azobenzens” check the title
Author Response

(The authors gave the same response as above.)

Reviewer 4 Report
The paper entitled “Photo-Isomerization Kinetic of Azobenzene Containing Surfactant Conjugated with Polyelectrolyte” deals with the effects of polymer and salt concentration, wavelength and irradiation intensity on the photo-isomerization kinetics of azobenzene containing surfactants complexed with poly(acrylic acid) sodium salt in aqueous solution.
The topic is interesting and supported by experimental sections. However, I suggest to replace Figure S1 in the supplementary materials because is exactly the same reported in Arya et al. J. Chem. Phys. 2020, 152, 024904 as Figure 1b. The text should be checked for minor tips (for example line 40 “photo-stationary state is of s fundamental importance”; line 67-68 “Figure 1a shows the chemical structure of azobenzene containing surfactant and the UV-Vis absorption spectra of the polymer-surfactant complex” could be replaced with “The chemical structure of azobenzene containing surfactant and the UV-vis absorption spectra of the polymer-surfactant complex are shown in Figure 1”; in the title "Kinetic" should be replaced by "Kinetics"; etc).
In conclusion I recommend the publication of the paper after minor revision in the light of the above remarks.
Author Response
Dear Reviewer,
Thank you very much for the comments and suggestions. In the following we answer to them:
- I suggest to replace Figure S1 in the supplementary materials because is exactly the same reported in Arya et al. Chem. Phys. 2020, 152, 024904 as Figure 1b.
Our answer:
The intention of the figure is to show the relative concentrations of trans and cis in the system (in both the papers). For the sake of removing any kind of overlaps, the figure in the current paper has been modified.
- The text should be checked for minor tips (for example line 40 “photo-stationary state is of s fundamental importance”
Our answer:
The typo has been corrected (see highlighted version of the manuscript).
- line 67-68 “Figure 1a shows the chemical structure of azobenzene containing surfactant and the UV-Vis absorption spectra of the polymer-surfactant complex” could be replaced with “The chemical structure of azobenzene containing surfactant and the UV-vis absorption spectra of the polymer-surfactant complex are shown in Figure 1”
Our answer:
We thank the referee for the suggestion. The former sentence has now been replaced by the suggested one.
- In the title "Kinetic" should be replaced by "Kinetics"; etc).
Our answer:
We thank the referee for the suggestion, and we updated the title.
Round 2
Reviewer 1 Report
The authors have answered my concerns. I recommend publishing the article. However, I have two points that I rise to the attention of the authors, changes to manuscript at their own discretion
1) Please consider revising the sentence "This begs to the suggestion that it applies to surfactant complexes with any polymers since surfactant is a surface active material and will always have a certain tendency to aggregate at interfaces.” This goes beyond what can be deduced of the data.
2) Additionally, I originally asked what is the pH. The authors answered this in the response but I strongly suggest putting the information also in the manuscript for the readers.
Reviewer 2 Report
The authors answered to the questions and did the corrections asked
Reviewer 3 Report
The authors followed all the suggestions after the first round of revision, improving the quality of the paper, which I will now gladly recommend for acceptance.